# Evaluation of Patients’ Perception of Safety in an Italian Hospital Using the PMOS-30 Questionnaire

**DOI:** 10.3390/ijerph18094515

**Published:** 2021-04-24

**Authors:** Sara Schiavone, Angela Annecchiarico, Danilo Lisi, Mario Massimo Mensorio, Francesco Attena

**Affiliations:** 1Department of Experimental Medicine, University of Campania “Luigi Vanvitelli”, 80138 Naples, Italy; sara.schiavone@unicampania.it; 2Azienda Ospedaliera Sant’Anna e San Sebastiano Caserta, 81100 Caserta, Italy; annecchiarico@gmail.com (A.A.); danilolisi@libero.it (D.L.); mariomassimo79@yahoo.it (M.M.M.)

**Keywords:** patient safety, hospital, questionnaire, PMOS-30, patient feedback, feasibility

## Abstract

Background: In our study, an Italian version of the PMOS-30 questionnaire was used to evaluate its feasibility and to improve health care quality in an Italian hospital. Methods: A cross-sectional study was conducted with 435 inpatients at a hospital in the Campania Region of Southern Italy using the PMOS-30 questionnaire and two other questions to assess patient feedback about the overall perception of safety. Results: The item “I was always treated with dignity and respect” showed the greatest percentage of agreement (agree/strongly agree = 89.2%; mean = 4.24). The least agreement was associated with the four “Staff Roles and Responsibilities” items (agree/strongly agree ranged from 31.5 to 40.0%; weighted mean = 2.84). All other 25 items had over 55.0% agreement, with 19 items over 70%. Moreover, 94.5% of the patients considered the safety of the ward sufficient/good/very good, and 92.8% did not notice situations that could cause harm to patients. Conclusion: Patient perception of safety was found to be satisfactory. The results were presented to the hospital decision makers for suggesting appropriate interventions. Our experience showed that the use of the PMOS-30 questionnaire may improve safety and health care quality in hospital settings through patient feedback.

## 1. Introduction

Patient safety is defined as the prevention of errors and adverse effects to patients associated with health care [1]. The importance of measuring patient safety in the health care setting has become widely recognized only in the past two decades. Therefore, before the 2000s, patient safety was not, with some exceptions [2], clearly incorporated into the dimensions of health care quality [3,4,5,6,7,8,9,10] and the agenda of health care organizations [11]. In 1984, Maxwell [3] identified the following dimensions of health care quality: access to service, relevance to need (for the whole community), effectiveness (for individual patients), equity (fairness), efficiency and economy, and social acceptability. This was followed by Donabedian’s widely cited 1990 paper [4] that outlined seven dimensions of quality: efficacy, effectiveness, efficiency, optimality, acceptability, legitimacy, and equity. Lastly, in 1998, 11 dimensions and 21 sub-dimensions of health care service quality were identified; however, there were no items explicitly about patient safety [8]. Only with the new century has a safety culture permeated into the values and norms of health care organization members [12,13,14]. Now, patient safety has become an important element of care quality, first evaluated by experts and health professionals and ultimately including patients’ feedback about their care [15,16,17].

In Italy, the Ministry of Health provided a variety of documents online about patient safety: recommendations for safety; reports on monitoring and analysis of sentinel events; reports on compliance from health care organizations with regional and national legislation; elaboration of many manuals and guidelines (about audit, root cause analysis, safety walk-around, safety in surgery, etc.) [18]. In recent years, several instruments and guidelines to improve patient safety have been developed using a patient-centred view [19]. Many of these evaluate patient feedback as a resource for improving patient safety in a hospital setting [20].

The Patient Measure of Safety (PMOS) is a recent structured questionnaire, designed to measure the patient’s perception of safety [17,20]. Some authors of this paper are currently validating an Italian version of the PMOS-30 questionnaire [21].

In the present study, this Italian version of the PMOS-30 questionnaire was used in a hospital setting for evaluating its feasibility in an Italian hospital and for promoting the improvement of health care quality. This was a case study for extending the use of the PMOS-30 questionnaire in a wider Italian context.

## 2. Methods

### 2.1. Setting and Participants

An epidemiological cross-sectional study was conducted in one general hospital in the Campania Region, Southern Italy, to assess patient feedback on safety using the PMOS-30 questionnaire. All wards were involved, except the intensive/sub-intensive care, psychiatric, and COVID-19 isolation wards. The medical researcher visited the hospital five days a week. With appropriate scheduling, each ward was investigated every three to five days during the data collection period, and patients who had been hospitalised for at least three days were included.

Inclusion criteria were a minimum age of 18 years and Italian language speaking. Exclusion criteria included cognitive impairment, severe psychiatric disease, and end-stage disease. Participants were asked to sign an informed consent form.

The questionnaire was self-administered, and the patients were informed that participation was voluntary and that they could withdraw from the study at any time with no consequences. The medical researcher who delivered the questionnaire, a resident in Public Health, Epidemiology, and Hospital Organization, was independent of the hospital staff and was available to answer participants’ questions about the questionnaire. He could not influence the patients, nor could he see what they were writing.

The time required to complete the questionnaire was about 15–20 min.

After compilation, the completed questionnaires were immediately placed in a strictly private folder by the medical researcher; therefore, the privacy of patients was ensured, and the answers remained confidential. Patients were informed that all data collected would be analysed and aggregated and that their confidentiality would be strictly protected. The health care professionals of the wards were not informed about the content of the questionnaire.

Data were collected between August 2020 and November 2020.

This study is a part of a research project conducted to validate an Italian version of the PMOS-30 questionnaire that will be published elsewhere [21]. In summary, the validation was carried out through confirmatory factor analysis and inter-item correlation. The English PMOS-30 questionnaire was translated into Italian and culturally adapted using standard forward–backward procedures performed by a multidisciplinary team [21].

Ethical approval for this study was obtained from the Ethics Committee of the University of Campania “Luigi Vanvitelli” (prot. N 0008664/i-2020).

### 2.2. Questionnaire

The questionnaire was divided into three sections. The first section contained an Italian version of the PMOS-30 questionnaire. The first version of the PMOS contained 44 items, but for the hospital setting, two shorter versions were created: PMOS-10 and PMOS-30 [22].

PMOS-30 included 30 items and 8 domains known to contribute to hospital safety: (1) communication and team working; (2) organisation and care planning; (3) access to resources; (4) ward type and layout; (5) information flow; (6) staff roles and responsibilities; (7) staff training; and (8) delays. All items were measured using a 5-point Likert scale (1—strongly disagree, 2—disagree, 3—neither disagree or agree, 4—agree, 5—strongly agree). There was also the option of “I prefer not to answer/I don’t know”. The second section consisted of two other items (not included in the PMOS-30) that investigated the patient’s overall perception of safety through two direct questions and one open answer question: “How do you rate the safety of this ward?” (5-point Likert scale: 1—very bad, 2—bad, 3—sufficient, 4—good, 5—very good); “Have you noticed any events that could have caused harm to patients?” (yes, no); “If yes, describe” (open answer). The third section collected socio-demographic data and hospital characteristics: gender (male, female), age (18–40, 41–55, 56–70, > 70), education level (primary school, middle school, high school, degree), marital status (married, unmarried, other), employment (employed, housewife, retired), nationality (Italian, not-Italian), ward type (open answer), and days of hospitalization (continuous).

### 2.3. Statistical Analysis

For each item of the Italian version of the PMOS-30 questionnaire, mean and standard deviations were calculated, and for each domain, the weighted mean and Cronbach alpha were calculated. Scores for items with negative questions were reversed (items 5, 8, 9, 11, 12, 16, 17, 18, 19, and 20). Therefore, the percentage of agreement refers to the opposite meaning of the questions. “I prefer not to answer/I don’t know”, and blank responses were treated as missing values.

The patients’ reported educational levels were standardised using the International Standard Classification of Education (ISCED 2011), which allows for cross-national comparisons of educational levels, and dichotomised into two groups: ISCED 0–2 (primary school, middle school) and ISCED 3–8 (high school, degree).

Several bivariate analyses were performed to determine if there were relationships between ward, days of hospitalization, and socio-demographic characteristics and some questionnaire items. Therefore, all the pertinent variables were dichotomized: ward type (medicine/surgery), days of hospitalization (3–5/ > 5), age (18–55/ > 55), marital status (married/other), employment (employed/housewife, retired), nationality (Italian/other), items of the PMOS-30 questionnaire (strongly disagree, disagree, neither agree or disagree/agree, strongly agree), and “How do you rate the safety of this ward?” (very bad, bad/sufficient, good, very good). Only associations <0.01 were considered statistically significant and presented in the results.

The sample size was estimated to be at least 400 subjects, assuming a 50% expected mean prevalence of “agree/strongly agree” in the PMOS-30 questionnaire, with precision of 5% and level of significance of 95%.

## 3. Results

### 3.1. Socio-Demographic Characteristics

A total of 474 inpatients were approached, and 39 declined to participate; therefore, the response rate was 91.8%. Of the 435 participants, 55.2% were male (Table 1).

The respondents ranged in age from 18 to 90 years, and almost a third (30.1%) were over 70 years old. Education level was equally distributed between the two groups: primary and middle school (54.0%), high school and degree (46.0%). Half of the participants were not formally employed. Only 15 inpatients were not Italian (3.4%); this was lower than the percentages recently reported for non-Italians residing in Campania (4.6%) and in Italy (8.7%) [23].

### 3.2. Questionnaire

Four items showed a percentage of agreement greater than 80% (Table 2): item 1 about “Dignity and Respect” (agree/strongly agree = 89.2%; mean = 4.24); item 18 about comfort of lighting levels (agree/strongly agree = 84.4%; mean = 3.84); items 27 and 28 of the domain 7, “Staff Training” (agree/strongly agree = 84.4–86.2%; weighted mean = 3.98).

The least agreement was found for the four items of domain 6, “Staff Roles and Responsibilities” (agree/strongly agree = 31.5–40.0%; weighted mean = 2.84). Question 8, “Staff didn’t always know when a doctor changed my plan of care”, had only 48.5% agreement (agree/strongly agree) and a high mean (3.49) because missing values were reported for 124 patients (28.5%). All the other domains and their 25 items showed agreement over 55.0%, with 19 items showing agreement over 70%. Conversely, except for domain 6, almost all (24/26) the other items did not exceed 17.0% of disagreement (disagree/strongly disagree). Many items garnered high level of the “I prefer not to answer/I don’t know” response and blank responses (>50 in case of items 8, 11, 12, 21, 22, and 29).

The results of the PMOS-30 questionnaire were then disaggregated for all the hospital and sociodemographic variables. In Table 3, only items with an association with *p* < 0.01 were included. Patients in the ISCED 0–2 group agreed more often than others with the eight items about safety (items 1, 2, 5, 6, 9, 17, 18, and 27). Conversely, they were less informed on three of the four items on “Staff Roles and Responsibilities” (items 23–25). Furthermore, patients under 55 years of age had better knowledge about all four items on “Staff Roles and Responsibilities” (items 23–26). In domains 1 and 6, Cronbach’s alpha was >0.8, in domains 3 and 4 it was >0.6, and in domain 2 it was 0.525.

Finally, the overall perception of the hospital’s safety was evaluated using two other questions that were not included in the PMOS-30 questionnaire: “How do you rate the safety of this ward?” and “Have you noticed any events that could have caused harm to patients?”. The majority (94.5%) of the patients considered the safety of the ward to be sufficient/good/very good, and 92.8% did not notice any situation that could cause harm to patients (Table 4).

Among the remaining 31 patients who reported potential harms, more than half of the reports were related to the ward staff (incompetence; poor attention, poor communication, and poor information to patients; delays in health care). Other answers concerned patient falls, structural deficiencies, excessive noise, and poor cleaning. There were no differences in the answers given by patients with different sociodemographic characteristics.

## 4. Discussion

In 2018, the OECD stated that the true extent of safety and harm across all health care settings is still a black box. Therefore, measuring safety is the starting point for improving patient safety because without measurement, actions to drive improvement are impossible [13].

We assessed patient feedback on safety using an Italian version of the PMOS-30 questionnaire. To date, the PMOS questionnaire has been validated in the Australian [24] and Persian settings [25]; this is the first time that the PMOS-30 questionnaire has been used in an Italian hospital setting.

Overall, the results of the PMOS-30 questionnaire showed satisfactory patient feedback on safety, considering that the lowest mean was 2.7 (item 24) and that all means were above 3.4, except in the “Staff Roles and Responsibilities” domain.

Patients agreed most often with the item “I was always treated with dignity and respect”. This result is in agreement with that obtained by Taylor et al. [24]. On the other hand, patients’ knowledge of “Staff Roles and Responsibilities” was the lowest. This result stems from a characteristic of many Italian health care professionals, an avoidance of being recognized; therefore, there is a low propensity to communicate one’s name to patients. For this reason, the Italian legislation decreed, in 1995, that patients must be informed about the identity of the health care staff, who must attach an identification tag to their gowns [26].

It is not surprising that there was a high number of “I prefer not to answer/I don’t know” and blank responses because there were many questions that patients might not have been able to answer. For example, the two items with the highest numbers of missing values (“I prefer not to answer/I don’t know” and blank responses) were “Staff didn’t always know when a doctor changed my plan of care” (*n* = 124) and “Information about me that my health care team needed was always available (e.g., drug charts, medical notes, test results)” (*n* = 98). These missing data probably resulted from poor communication between health care professionals and patients, so patients might not have asked about these interactions, or not have felt empowered to ask about these interactions. This aspect of communication is another topic often emphasized in continuing education courses for health care professionals. Better communication between health care professionals and patients leads to better outcomes with less cost and work efforts [27,28].

The overall perception of safety (determined using two questions not included in the PMOS-30 questionnaire) was satisfactory. Only 24 patients judged safety as bad/very bad. Only 31 identified events that could have caused harms, more than half of which concerned ward staff behaviours, among which poor attention, communication, and information to patients prevailed. These data align with the previous results that highlighted the issue of poor communication between health care professionals and patients [28].

The results of our study were presented to the hospital decision makers with the aim of suggesting interventions to improve safety in those areas where the less satisfactory results were obtained. The main intervention was to improve the relationship and communication between health care professionals and patients. The decision makers were also presented with a list of the 31 potential harms reported by patients to allow selective interventions in individual wards. The hospital health manager also shared the results of the study with the heads of the various selected wards to make it easier for the heads to recognize weaknesses and take effective corrective action. Therefore, with this instrument, targeted improvements have been made in individual wards to increase overall patient safety.

The study had the following limitations. This was the first experience with the PMOS-30 questionnaire within a single Italian hospital. While satisfactory results for improving patients’ safety have been achieved, the instrument needs to be applied in a wider context to better assess its feasibility. Moreover, questionnaire administration to inpatients can create anxiety in the patients about reporting undesirable results because those may become known by the staff. Consequently, this could reduce reporting of negative answers. Therefore, many measures were put in place to reduce patient anxiety about this occurrence, as described in the Methods section. However, a residual fear cannot be ruled out. Regarding the question out of the PMOS-30 questionnaire (“Have you noticed any events that could have caused harm to patients?”), we have not informed the participants which events could lead to harm. Therefore, the answers reporting potential harm to patients might have been underestimated.

## 5. Conclusions

We believe that, if the hospital decision makers are involved, the use of the PMOS-30 questionnaire might improve safety and health care quality in hospital settings through patient feedback. Future research studies with the routine use of the PMOS-30 questionnaire in a hospital setting might highlight gaps and deficiencies in individual wards and allow focused improvement.

## Figures and Tables

**Table 1 ijerph-18-04515-t001:** Socio-demographic and hospital characteristics.

Socio-Demographic Characteristics		*n*	%
Gender	Male	240	55.2
	Female	194	44.6
	Missing	1	0.2
	Total	435	100
Age	18–40	98	22.5
	41–55	79	18.2
	56–70	125	28.7
	>70	131	30.1
	Missing	2	0.5
	Total	435	100
Education	Primary school	91	20.9
	Middle school	144	33.1
	High school	127	29.2
	Degree	73	16.8
	Total	435	100
Marital status	Married	307	70.5
	Unmarried	72	16.6
	Other	56	12.9
	Total	435	100
Employment	Employed	192	44.1
	Unemployed	13	3.0
	Housewife	68	15.6
	Retired	156	35.9
	Missing	6	1.4
	Total	435	100
Nationality	Italian	420	96.6
	Not Italian	15	3.4
	Total	435	100
Hospital characteristics			
Ward	Medicine	225	51.7
	Surgery	208	47.8
	Missing	2	0.5
	Total	435	100
Days of hospitalization	3–5	218	50.1
	>5	216	49.7
	Missing	1	0.2
	Total	435	100

**Table 2 ijerph-18-04515-t002:** Patients’ perception of safety with PMOS-30 questionnaire.

Items	*n*	Agree	Disagree	Missing	Mean	SD
Dignity and respect						
1. I was always treated with dignity and respect	431	89.2%	3.2%	4	4.24	0.753
Communication and team working (domain 1)						
2. I got answers to all the questions I had about my care	428	77.5%	8.5%	7	3.86	0.828
3. I always felt staff listened to me about my concerns	424	73.8%	11.3%	11	3.75	0.887
4. There was always someone available to deal with every aspect of my care	425	71.7%	11.3%	10	3.75	0.902
5. I felt that the attitude of staff towards me was poor (R)	427	77.9%	11.7%	8	3.80	0.889
6. Staff worked together as a team here	403	67.6%	14.3%	32	3.66	1.010
Weighted mean					3.76	
Cronbach’s alpha: 0.828						
Organization and care planning (domain 2)						
7. My medicines were always available	415	73.1%	15.6%	20	3.77	0.974
8. Staff didn’t always know when a doctor changed my plan of care (R)	311	48.5%	17.0%	124	3.49	1.034
9. Staff gave me conflicting information about my care (R)	406	75.2%	11.5%	29	3.76	0.870
10. When I needed treatment, there was always someone available who was trained to do it	418	75.9%	9.4%	17	3.82	0.871
Weighted mean					3.72	
Cronbach’s alpha: 0.525						
Access to resources (domain 3)						
11. Staff/patients waited a long time for porters to arrive (R)	369	67.8%	10.3%	66	3.74	0.873
12. Staff seemed to struggle to get help when they needed it (R)	384	65.7%	14.9%	51	3.66	0.932
13. Equipment and supplies were always available when needed (e.g., hoists, bed pans, walking aids, dressings)	396	79.5%	5.1%	39	3.94	0.680
Weighted mean					3.78	
Cronbach’s alpha: 0.634						
Ward type and layout (domain 4)						
14. Staff were prompt in answering my buzzer	408	74.5%	6.7%	27	3.99	0.816
15. The ward was able to deal with all my treatment needs	414	77.0%	10.1%	21	3.84	0.843
16. Lack of space made it difficult for staff to do their jobs (R)	412	75.6%	13.8%	23	3.75	0.890
17. The following aspects of the ward made it uncomfortable for me: Noise levels (R)	424	63.9%	23.4%	11	3.46	1.095
18. The following aspects of the ward made it uncomfortable for me: Lighting levels (R)	423	84.4%	9.0%	12	3.84	0.720
19. The following aspects of the ward made it uncomfortable for me: Temperature (R)	427	74.3%	17.0%	8	3.63	0.932
20. The following aspects of the ward made it uncomfortable for me: Poor cleanliness (R)	421	79.8%	9.2%	14	3.93	0.868
Weighted mean					3.77	
Cronbach’s alpha: 0.674						
Information flow (domain 5)						
21. Information about me that my healthcare team needed was always available (e.g., drug charts, medical notes, test results)	337	64.6%	6.4%	98	3.83	0.782
22. After shift changes, staff knew important information about my care	382	74.5%	6.7%	53	3.92	0.762
Weighted mean					3.87	
Cronbach’s alpha: N/A						
Staff roles and responsibilities (domain 6)						
23. I knew what the different roles of the people caring for me were	411	39.8%	46.0%	24	2.89	1.190
24. It was clear who was in charge of the ward staff	416	31.5%	68.6%	19	2.70	1.154
25. I knew which consultant was in charge of my care	422	40.0%	51.5%	13	2.89	1.191
26. I always knew which nurse or nurses were responsible for my care	419	37.0%	48.3%	16	2.89	1.108
Weighted mean					2.84	
Cronbach’s alpha: 0.824						
Staff training (domain 7)						
27. Staff were always able to use the necessary equipment	402	84.4%	3.7%	33	4.00	0.607
28. Staff were always able to carry out tasks that they should be able to do	426	86.2%	4.6%	9	3.98	0.651
Weighted mean					3.98	
Cronbach’s alpha: N/A						
Delays (domain 8)						
29. There were enough staff on the ward to get things done on time	379	56.1%	23.9%	56	3.41	1.069
30. My treatment/procedure/operation always happened on time	425	71.7%	13.6%	10	3.67	0.906
Weighted mean					3.54	
Cronbach’s alpha: N/A						

Agree = agree/strongly agree; Disagree = disagree/strongly disagree; Missing = “I prefer not answer/I don’t know” and blank responses; N/A = not applicable; SD = standard deviation; (R) = reverse questions.

**Table 3 ijerph-18-04515-t003:** Hospital and socio-demographic characteristics with higher agreement.

Items *	ISCED 0–2 vs. ISCED 3–8	Medicine vs. Surgery	18–55 vs. >55
	*p* Value < 0.01		
Dignity and respect			
1. I was always treated with dignity and respect	0.004		
Communication and team working (domain 1)			
2. I got answers to all the questions I had about my care	0.005		
5. I felt that the attitude of staff towards me was poor	0.003	0.003	
6. Staff worked together as a team here	0.009	0.006	
Organization and care planning (domain 2)			
7. My medicines were always available			0.001
9. Staff gave me conflicting information about my care	0.005		
Ward type and layout (domain 4)			
17. The following aspects of the ward made it uncomfortable for me: Noise levels	0.000		
18. The following aspects of the ward made it uncomfortable for me: Lighting levels	0.001		
Staff roles and responsibilities (domain 6)			
23. I knew what the different roles of the people caring for me were	0.002 (R)		0.000
24. It was clear who was in charge of the ward staff	0.000 (R)		0.000
25. I knew which consultant was in charge of my care	0.000 (R)		0.000
26. I always knew which nurse or nurses were responsible for my care			0.000
Staff training (domain 7)			
27. Staff were always able to use the necessary equipment	0.004		

(R) = reverse agreement (i.e., ISCED 3–8 vs. ISCED 0–2); * strongly disagree, disagree, neither agree or disagree/agree-strongly agree.

**Table 4 ijerph-18-04515-t004:** Overall patient perception of safety.

How do you rate the safety of this ward?		
	*n*	%
Very Bad	6	1.4
Bad	18	4.1
Sufficient	111	25.5
Good	208	47.8
Very Good	91	21.0
*Missing*	1	0.2
Total	435	100
Mean = 3.83; SD = 0.853		
Have you noticed any events that could have caused harm to patients?		
	*n*	%
Yes	31	7.1
No	401	92.2
*Missing*	3	0.7
Total	435	100

SD = standard deviation.

## Data Availability

The data presented in this study are available on request from the corresponding author.

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
