# Peer review of "Evaluation of Patients’ Perception of Safety in an Italian Hospital Using the PMOS-30 Questionnaire"

_ijerph, 2021, doi:10.3390/ijerph18094515_

Round 1
Reviewer 1 Report
The general topic is very interesting and the paper is well written. The study is an evaluation of the PMOS-30 questionnaire feasibility in an Italian hospital. As such, the contribution of the research is very limited. The manuscript just presents a summary of the results of the questionnaire. However, no further analysis was presented. Originality and novelty are not observed in the manuscript. There is not an instrument development or model evaluation to investigate the constructs of the survey. In addition, as authors stated, the instrument needs to be applied in a wider context to better assess its feasibility.
Therefore, the scientific soundness of the manuscript is low, providing few information to readers and researchers.
Author Response
Response to Reviewer 1 Comments
COMMENT: The general topic is very interesting and the paper is well written. The study is an evaluation of the PMOS-30 questionnaire feasibility in an Italian hospital. As such, the contribution of the research is very limited. The manuscript just presents a summary of the results of the questionnaire. However, no further analysis was presented. Originality and novelty are not observed in the manuscript. There is not an instrument development or model evaluation to investigate the constructs of the survey. In addition, as authors stated, the instrument needs to be applied in a wider context to better assess its feasibility. Therefore, the scientific soundness of the manuscript is low, providing few information to readers and researchers.
REPLY: The study presents a subsequent evaluation of the Italian version of the PMOS-30 questionnaire. A previous validation by confirmatory factor analysis after standard forward-backward procedures of translation was carried out by some of the authors of this manuscript and is under revision, as cited line 55-56,83-84. Therefore, the methods and results are not reported in the present study. In Italy, the issue of patient safety is very important but there are no evaluation instruments from the patients' point of view. This instrument represents a novelty in Italy and could be the beginning of its use in a wider context as in England and Australia. In addition, further improvements were made in response to other reviewers.
Reviewer 2 Report
The manuscript describes the testing of an Italian version of the PMOS30 questionnaire, as well as some additional satisfaction and demographic questions. while overall the methods were simply to understand, the description of the method for negative questions in the survey being 'reversed' was not clear. Were the questions 'reversed' or just the scores? the way these are reported in the results/tables is misleading and could be made clearer.
I dislike that the authors use the sexist term 'housewife' in their data collection and reporting, clearly making the assumption the only people in 'home duties' roles are female. Also within the same data collection category they did not provide the options 'unemployed': were these people excluded, were none recruited, was the option never provided on the survey? Likewise the only options for gender are the binary 'female/male' options with no option reported for non-binary people. Again, was this even asked on the survey, or just not reported?
The repetition of the demographic survey options both in the text and in the table is unnecessary.
In the results, the authors say "of the 43 participants". Which 43 participants? I suspect this is a typing error and meant to be 434?
The use of 'n." in the reporting of results for the questionnaire is not appropriate grammar and can be removed throughout.
The use of the term affirmative terms such as 'agreement' when discussing a negative statement "staff didn't know when a doctor ..." is confusing. Improving the grammar and sentence structure of the sentences reporting outcomes for the questionnaire statements should be improved, perhaps to remove the use of quotation marks entirely.
The questionnaire asked patients about 'events that could have caused harm'. However, patients can have limited understanding about what can and cannot cause harm, and the authors fail to discuss this. Did the authors provide participants with information about what events can lead to harm?
The authors also asked participants about their perception of "staff didn't always know when a doctor changed my plan" and note this had the highest nonresponse rate. However, the authors do not explore the reasons why there is such a high nonresponse to this question compared to others. This warrants consideration in the manuscript, e.g. patients might not observe these interactions, might not ask about about these interactions, or not feel empowered to ask about these interactions due to the clinician-patient imbalance.
Please provide evidence for the statements about Italian health care professionals refraining from sharing their name with patients.
Please provide a reference for the sentence "these data align with the previous restuls ..."
Author Response
Response to Reviewer 2 Comments
COMMENT: The manuscript describes the testing of an Italian version of the PMOS30 questionnaire, as well as some additional satisfaction and demographic questions. while overall the methods were simply to understand, the description of the method for negative questions in the survey being 'reversed' was not clear. Were the questions 'reversed' or just the scores? the way these are reported in the results/tables is misleading and could be made clearer.
REPLY: we clarified the meaning of “reversed”.
NEW VERSION line 114-116: Scores for items with negative questions were reversed (Items 5, 8, 9, 11, 12, 16, 17, 18, 19 and 20). Therefore, the percentage of agreement refers to the opposite meaning of the questions.
COMMENT: I dislike that the authors use the sexist term 'housewife' in their data collection and reporting, clearly making the assumption the only people in 'home duties' roles are female. Also within the same data collection category they did not provide the options 'unemployed': were these people excluded, were none recruited, was the option never provided on the survey? Likewise the only options for gender are the binary 'female/male' options with no option reported for non-binary people. Again, was this even asked on the survey, or just not reported?
REPLY: Thank you for your comment, we must correct Table 1. The item “employed” in Table 1 arised from the open question “profession” in the questionnaire. Therefore, 68 women filled in “housewife”. For a mistake we did not included in the Table 1, 13 unemployed participants.
Moreover, we guiltily admit to having included in the question only the binary 'female/male' options.
NEW VERSION line 140: new table.
COMMENT: The repetition of the demographic survey options both in the text and in the table is unnecessary.
REPLY: we revised the comments on demographic survey options.
NEW VERSION: line 142-147: The respondents ranged in age from 18–90 years, and almost a third (30.9%) were over 70 years old. Education level was equally distributed between two groups: primary and middle school (54.0%), high school and degree (46.0%). Half of the participants were not formally employed. they were namely housewives (15.8%) and retired (36.4%). Most of the participants were married (70.6%). Only 15 inpatients were not Italian (3.4%); this is lower than the percentages recently reported for non-Italians residing in Campania (4.6%) and in Italy (8.7%) [23].
COMMENT: In the results, the authors say "of the 43 participants". Which 43 participants? I suspect this is a typing error and meant to be 434?
REPLY line 138: corrected, they were 435.
COMMENT: The use of 'n." in the reporting of results for the questionnaire is not appropriate grammar and can be removed throughout.
REPLY: we eliminated all the “n.”
NEW VERSION line 150-158.
COMMENT: The use of the term affirmative terms such as 'agreement' when discussing a negative statement "staff didn't know when a doctor ..." is confusing. Improving the grammar and sentence structure of the sentences reporting outcomes for the questionnaire statements should be improved, perhaps to remove the use of quotation marks entirely.
REPLY: we used the same criteria of the original application of PMOS questionnaire. See also reply to your first comment.
NEW VERSION line 114-116: Scores for items with negative questions were reversed (Items 5, 8, 9, 11, 12, 16, 17, 18, 19 and 20). Therefore, the percentage of agreement refers to the opposite meaning of the questions.
COMMENT: The questionnaire asked patients about 'events that could have caused harm'. However, patients can have limited understanding about what can and cannot cause harm, and the authors fail to discuss this. Did the authors provide participants with information about what events can lead to harm?
REPLY: we did not provide participants with information about what events can lead to harm. Thank you for this comment, so we can include this consideration in the ‘limitation’ section.
NEW VERSION line 249-252: Regarding the question out of PMOS-30 questionnaire (‘have you noticed any events that could have caused harm to patients?’) we have not informed the participants which events could lead to harm. Therefore, the answers reporting potential harm to patients might be underestimated.
COMMENT: The authors also asked participants about their perception of "staff didn't always know when a doctor changed my plan" and note this had the highest nonresponse rate. However, the authors do not explore the reasons why there is such a high nonresponse to this question compared to others. This warrants consideration in the manuscript, e.g. patients might not observe these interactions, might not ask about about these interactions, or not feel empowered to ask about these interactions due to the clinician-patient imbalance.
REPLY: In the paper we said (lined 225) that better communication between health care professionals and patients leads to better outcomes so these missing data probably depend on poor communication between healthcare professionals and patients. Consequently, we could clarify that patients might not ask about these interactions, or not feel empowered to ask about these interactions due to the clinician-patient imbalance.
NEW VERSION line 222-224: These missing data probably depend on poor communication between healthcare professionals and patients, so patients might not ask about these interactions, or not feel empowered to ask about these interactions.
COMMENT: Please provide evidence for the statements about Italian health care professionals refraining from sharing their name with patients.
REPLY: it is difficult to provide direct evidence about this argument, but as cited (line 213) Italian legislation decreed, since 1995, that patients must be informed about the identity of the health care staff who must attach an identification tag to their gown.
COMMENT: Please provide a reference for the sentence "these data align with the previous results ..."
REPLY: we revised the sentence and add reference.
NEW VERSION line 232-234: These data align with the previous results that highlighted the issue of health care professionals not communicating their names to patients poor communication between healthcare professionals and patients [28].
Reviewer 3 Report
This paper concerns patient satisfaction with care, which is very important. I have some questions.
The questionnaire is presented in the Introduction. This should be in the Methods section.
The aim for the study is unclear, needs to be revised. Using a questionnaire can not be a way to improve care quality, much more needs to be done.
Method: to investigate each ward every three to five days, how much is that? A visit every day? Needs to be written more clear.
How was the researcher available to the patients to answer questions? How about the ethics here? By whom were the questionnaires put in a private folder and where, at the ward? Could the patients respond in a private room?
Table 1: Line 129, were there only 43 participants?
The total number of participants differ. Why? Where are the missing patients?
Limitations: Was the q. to wide in structure? it takes time to fill in for the patient. Could the patients be sure of that staff did not see what they were writing? How were staff informed about this study?
Conclusion: too certain, how can safety be improved by this questionnaire? was there talks with the staff after?
Author Response
Response to Reviewer 3 Comments
COMMENT: The questionnaire is presented in the Introduction. This should be in the Methods section.
REPLY: we moved the presentation of questionnaire in the methods section
NEW VERSION: see line 94-96.
COMMENT: The aim for the study is unclear, needs to be revised. Using a questionnaire can not be a way to improve care quality, much more needs to be done.
REPLY: we agree that a questionnaire alone cannot be a way to improve care quality (and in the discussion we explained what we done to improve care quality). Therefore, we mitigated the aim of the study.
NEW VERSION line 58-59: In the present study, this Italian version of PMOS-30 questionnaire was used in a hospital setting for evaluating its feasibility and to improve the health care quality in an Italian hospital and for promoting the improvement of health care quality. This was a case study for extending the use of the PMOS-30 questionnaire into a wider Italian context.
COMMENT: Method: to investigate each ward every three to five days, how much is that? A visit every day? Needs to be written more clear.
REPLY: the researcher visited the hospital five day a week. With appropriate scheduling, the wards were visited no earlier than three days (so as not to find the same patients from the previous visit) and no more than 5 days (so as not to lose new patients)
NEW VERSION line 67-68: The medical researcher visited the hospital five day a week. With appropriate scheduling, each ward was investigated every three to five days during the data collection period, and patients who had been hospitalised for at least three days were included.
COMMENT: How was the researcher available to the patients to answer questions? How about the ethics here? By whom were the questionnaires put in a private folder and where, at the ward? Could the patients respond in a private room?
REPLY: During the completion of the questionnaire, the patient remained in his room. The researcher was at such a distance that he could not influence the patient or be able to see what he was writing. However, he was available for any clarification. In addition, he had a folder with him which was large enough to be able to anonymously introduce all the questionnaires of the day.
NEW VERSION line 78-83: He could not influence the patients or be able to see what they were writing. The time required to complete the questionnaire was about 15-20 minutes.
After compilation, the completed questionnaires were immediately placed in a strictly private folder by the medical researcher; therefore, the privacy of patients was ensured, and their answers remained confidential.
COMMENT: Table 1: Line 129, were there only 43 participants?
REPLY line 138: corrected, they were 435.
COMMENT: The total number of participants differ. Why? Where are the missing patients?
REPLY: IN Table 1 we added, for each item, the missing values.
NEW VERSION line 140: we added missing values in Table 1.
COMMENT: Limitations: Was the q. to wide in structure? it takes time to fill in for the patient. Could the patients be sure of that staff did not see what they were writing? How were staff informed about this study?
REPLY: The patients took about 15-20 minutes to fill in the questionnaire. They never manifested any time problems in completing it, as the chosen time was after the medical examination round and away from mealtimes. The only hospital employees informed about the content of the questionnaire were the health administrators.
NEW VERSION line 80-86: The time required to complete the questionnaire was about 15-20 minutes. After compilation, the completed questionnaires were immediately placed in a strictly private folder by the medical researcher; therefore, the privacy of patients was ensured, and their answers remained confidential. Patients were informed that all data collected would be analysed and aggregated and that their confidentiality would be strictly protected. The healthcare professionals of the wards were not informed about the content of the questionnaire.
COMMENT: Conclusion: too certain, how can safety be improved by this questionnaire? was there talks with the staff after?
REPLY: we mitigated the conclusion
NEW VERSION line 252-254: This study has shown We believe that, if the hospital decision makers are involved, the use of the PMOS-30 questionnaire may might improve safety and health care quality in hospital settings through patient feedback. Future research studies with the routine use of the PMOS-30 questionnaire in a hospital setting might highlight gaps and deficiencies in individual wards and allow focused improvement.
Round 2
Reviewer 1 Report
Thank you for the opportunity to revise the edited version of the manuscript.
The study has minor changes from the previous version. As such, the manuscript does not present a statistical analysis on the constructs (e.g. factor analysis) neither model evaluation. Instead, authors reserve the factor analysis for another manuscript submitted elsewhere, thus, the scientific soundness of the manuscript is low.
Moreover, originality and novelty are not observed in the manuscript; presenting a partial analysis of a cross sectional study of POMS-30 in a hospital, might be of limited interest for readers and researchers.
Finally, as authors stated, “an Italian version of the PMOS-30 questionnaire was used to evaluate its feasibility and to improve the health care quality in an Italian hospital”.
However, as the same authors argued in the discussion section “the instrument needs to be applied in a wider context to better assess its feasibility”. Regarding the feasibility, which is one of the main goals, this is not determined in the research. Moreover, regarding “to improve the health care quality”, authors conclude “the use of the PMOS-30 questionnaire may might improve safety and health care quality in hospital settings through patient feedback”, thus, the manuscript does not show any evidence regarding both, the feasibility and improving the quality of care.
With the aforementioned, the overall contribution of the research is very limited.
Author Response
Response to Reviewer 1 Comments
COMMENT: Thank you for the opportunity to revise the edited version of the manuscript.
The study has minor changes from the previous version. As such, the manuscript does not present a statistical analysis on the constructs (e.g. factor analysis) neither model evaluation. Instead, authors reserve the factor analysis for another manuscript submitted elsewhere, thus, the scientific soundness of the manuscript is low.
REPLY: Thank you for your second revision. We integrated in the Methods section some information about the Italian validation of the PMOS-30 questionnaire.
NEW VERSION line 88-92: This study is a part of a research project that is being conducted to validate an Italian version of the PMOS-30 questionnaire and that will be published elsewhere [22]. In summary, the validation was carried out through confirmatory factor analysis and inter-item correlation. The English PMOS-30 questionnaire has been translated into Italian and culturally adapted using standard forward-backward procedures performed by a multidisciplinary team [22].
COMMENT: Moreover, originality and novelty are not observed in the manuscript; presenting a partial analysis of a cross sectional study of POMS-30 in a hospital, might be of limited interest for readers and researchers.
REPLY: Taking note of your comment, we think a first application in Italy, even if only in one main hospital, of a new instrument potentially capable to improve the quality of health care is an original contribution.
COMMENT: Finally, as authors stated, “an Italian version of the PMOS-30 questionnaire was used to evaluate its feasibility and to improve the health care quality in an Italian hospital”.
However, as the same authors argued in the discussion section “the instrument needs to be applied in a wider context to better assess its feasibility”. Regarding the feasibility, which is one of the main goals, this is not determined in the research. Moreover, regarding “to improve the health care quality”, authors conclude “the use of the PMOS-30 questionnaire may might improve safety and health care quality in hospital settings through patient feedback”, thus, the manuscript does not show any evidence regarding both, the feasibility and improving the quality of care.
With the aforementioned, the overall contribution of the research is very limited.
REPLY: In our opinion, the two statements are compatible. The first, (“an Italian version of the PMOS-30 questionnaire was used to evaluate its feasibility and to improve the health care quality in an Italian hospital”) refers to the feasibility in one main hospital, and we believe that this goal has been achieved. The second statement refers to a subsequent evaluation on a broader context.
Regarding “to improve the health care quality” we integrated the discussion section.
NEW VERSION line 242-245: The hospital health manager also shared the results of the study with the heads of the various selected wards to make it easier for the heads to recognize weaknesses and take effective corrective action. Therefore, with this instrument targeted improvements have been made in individual wards to increase overall patient safety.